# Distribution Consistency Guided Hashing for Cross-Modal Retrieval

## ABSTRACT

With the massive emergence of multi-modal data, cross-modal retrieval (CMR) has become one of the hot topics. Thanks to fast retrieval and efficient storage, cross-modal hashing (CMH) provides a feasible solution for large-scale multi-modal data. Previous CMH methods always directly learn common hash codes to fuse different modalities. Although they have obtained some success, there are still some limitations: 1) These approaches often prioritize reducing the heterogeneity in multi-modal data by learning consensus hash codes, yet they could sacrifice modality-specific information. 2) They frequently utilize pairwise similarities to guide hashing learning and neglect class distribution correlations. To overcome these two issues, we propose a novel Distribution Consistency Guided Hashing (DCGH) framework. Specifically, we first learn the modality-specific representation to extract the private discriminative information. Further, we learn consensus hash codes from the private representation by consensus hashing learning, thereby merging the specifics with consistency. Finally, we propose distribution consistency learning to guide hash codes following a similar class distribution principle between multi-modal data, thereby exploring more consistent information. Lots of experimental results on four benchmark datasets demonstrate the effectiveness of our DCGH on both fully paired and partially paired CMR tasks.

## CCS CONCEPTS

• **Information systems** → **Multimedia and multimodal retrieval**.

## KEYWORDS

Cross-modal retrieval, class center consistency, hashing learning, modality-specific learning.

## 1 INTRODUCTION

WITH the rapid development of social media and networks, the volume of multimedia data has experienced a massive expansion, typically encompassing diverse cross-modal data types like text, images, and videos. Such a swift proliferation of multimedia data highlights the pressing necessity for the efficient storage and retrieval of these extensive volumes of data. Therefore, cross-modal retrieval (CMR) [25, 35, 37, 42] is gaining significant attention as a fundamental task for retrieving semantically correlated instances across various query modalities. Recently, a considerable body of research on CMR has advocated for the acquisition of a unified, real-valued representation that encompasses all modalities, thereby mitigating the heterogeneous disparities in multi-modal data. While these methods have shown effectiveness, they compromise query and training efficiency. Hence, the new challenge lies in how to accurately and efficiently search for semantic similarities across modalities.

Owing to the great advantages of fast query and efficient storage for extensive multi-modal data, the hashing techniques [17, 24, 40] provide a feasible solution to apply in large-scale CMR. Therefore, many cross-modal hashing (CMH) methods have been proposed, whose key idea is to project cross-modal data into the Hamming space as binary hash codes while preserving the inherent semantic similarities. Then, the similarities between different instances can be calculated as Hamming distances through the XOR operation [20], resulting in minimal computational costs. The existing CMH methods can be broadly classified into two main types: supervised and unsupervised. As for supervised ones, some methods directly use label information to guide hash learning [7, 19, 22, 34], while others firstly use labels to construct similarity and then proceed to learn hash codes [16, 27, 28, 39]. For unsupervised CMH, some methods capture the structural similarity through graph learning [30, 36, 38], while others use matrix factorization to capture semantic similarity [2, 5, 26]. Generally, supervised CMH outperforms unsupervised ones by leveraging semantic information.

Although existing supervised methods have achieved significant progress, they still exhibit certain limitations that need to be addressed: (1) These methods tend to eliminate the heterogeneous gap of multi-modal data by learning the consensus hash codes, which leading to sacrifice modality-specific information from each modality; (2) They often rely on pairwise similarities to learn hash codes and ignore the spatial distribution correlations, which could not be able to fully mine consensus class information.

To address the aforementioned issues, we propose a Distribution Consistency Guided Hashing framework (DCGH) for cross-modal retrieval. DCGH considers that each modality-specific representation should obey the unified prior class information. In other words, they should enjoy the same class centers as the original multi-modal data. The basic framework of DCGH is depicted in Fig.1. Specifically, our DCGH first learns the modality-specific representation to extract the modality-specific discriminative information from each modality, thereby improving the quality of each private property to some extent. Then, DCGH adopts consensus hashing learning to represent the consistent structural information of all modalities. Finally, based on the principle of class center consistency between multi-modal data, DCGH proposes a distribution consistency learning strategy that enables dual representations (*i.e.*, modality-specific representation and consensus hash codes) to follow a similar class distribution, thereby exploring more consistent information. We summarize the following contributions from this paper:

- We propose a novel distribution consistency guided hashing for CMR. To our best knowledge, it is the first to simultaneously investigate modality-specific and distribution consistency learning, which enables a comprehensive exploration of the inherent correlations between all modalities.
- In order to promote the modality-specific discriminative representations to more realistically reflect the data distribution of all modalities, we propose distribution consistency learning that makes each specific representation obey the same class centers as the original data.
- Extensive experiments verify that our DCGH has better performance than some state-of-the-art comparison methods on four widely used datasets.

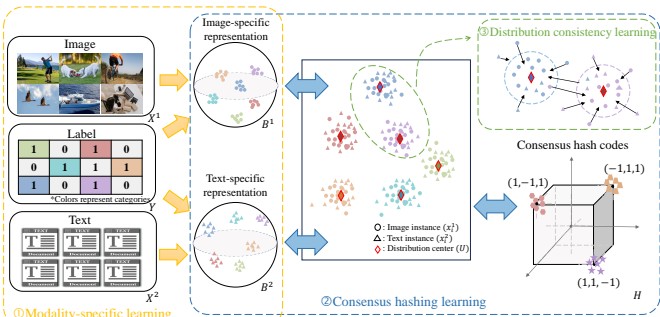

**Figure 1: The overall framework of our DCGH. The modality-specific representations are first learned from different modalities to extract each private property. Then, we learn common hash codes by consensus hashing learning, thereby inheriting the intrinsic specific information from each modality. Finally, we propose distribution consistency learning to endow the consistency of the class center between the representations for aggregating iteratively hash codes around each class center.**

## 2 RELATED WORK

### 2.1 Unsupervised Cross-modal Hashing

Since unsupervised CMH does not rely on the label semantic information provided by the original labels, it typically learns hash codes by exploring the structure similarities of the raw data. It can typically be categorized into matrix factorization-based methods and graph-based methods. To better explore the inherent structure of modalities, latent structure discrete hashing factorization (LSDHF) [5] extracts hash codes by aligning the eigenvalues of the similarity matrix. Regarding graph-based methods, adaptive structural similarity preserving hashing (ASSPH) [11] proposes an asymmetric semantic learning approach and uses graph-based structure to measure similarity. To better evacuate and preserve the intrinsic cross-modal semantics, correlation-identity reconstruction hashing (CIRH) [41] constructs a cross-modal collaborative graph to model heterogeneous cross-modal correlations. Considering the potent feature extraction and pattern analysis ability of deep neural networks (DNNs), there is a continuous emergence of unsupervised CMH methods based on DNNs. For instance, deep adaptively-enhanced hashing (DAEH) [18] proposes an adaptively-enhanced

optimization strategy and to learn hash functions. To address the performance degradation caused by hash binarization optimization, unsupervised contrastive cross-modal hashing (UCCH) [6] proposes a framework that performs hash optimization in contrastive learning. Although unsupervised CMH generally achieves satisfactory performance, they still exhibit a noticeable gap compared with supervised CMH.

### 2.2 Supervised Cross-modal Hashing

Supervised CMH, which uses semantic information provided by labels to guide hashing learning, usually performs better than unsupervised CMH. To model the potential manifold structure between heterogeneous data, label guided discrete hashing (LGDH) [9] simultaneously learns the manifold structure and hash codes with the guidance of labels. In order to bridge the semantic loss in the hashing process, scalable pairwise embedding constraint hashing (SPECH) [31] measures the semantic correlation of cross-modal data by using paired data to calculate the loss of likelihood similarity. More recently, asymmetric learning strategies have also been introduced into the CMH field. For instance, scalable asymmetric discrete cross-modal hashing (BATCH) [29] uses distance minimization to embed semantic information into hash codes, and an asymmetric strategy is proposed to bridge the gaps between the shared space and the hash space. Aimed at investigating the potential relevance of multi-label semantics, adaptive label correlation based asymmetric cross-modal hashing (ALECH) [10] proposes an asymmetric strategy to connect different feature spaces and adaptively learn latent features utilizing higher-order semantic labels. To handle the intrinsic correlations across modalities, asymmetric supervised fusion-oriented hashing (ASFOH) [32] establishes correlations between the shared latent representations and the semantic label matrix. Moreover, deep-based methods such as attention-aware deep cross-modal Hashing (TEACH) [33], which designs modality-specific attention modules for learning hash codes, also demonstrate impressive performance.

In general, existing CMH methods typically project multi-modal data into shared hash codes. Although these methods achieve promising performance, most of them overlook the fact that features from each modality contain both common semantics and modality-specific semantics. Additionally, they always rely on pairwise similarities to promote hashing learning. More importantly, they have yet to consider the preservation of class distribution consistency. To this end, we propose distribution consistency guided hashing to enhance both the expression and discriminative abilities of hash codes.

## 3 PROPOSED METHOD

### 3.1 Notations

In this paper, we denote the number of instances, classes, and modalities as $n$, $c$, and $m$, respectively. Then, multi-modal data can be represented as $O^v = [o_1^v, o_2^v, \cdots, o_n^v] \in \mathbb{R}^{h^v \times n}$ ($v = 1, 2, \cdots, m$), where $h^v$ is the feature dimensionality of the $v$-th modality. The corresponding labels are $Y \in \{0, 1\}^{c \times n}$. To be specific, if the $j$-th instance comes from the $i$-th class, its label is $Y_{ij} = 1$, otherwise $Y_{ij} = 0$. We adopt $\ell_2$-norm to normalize the i-th column

label, $i.e.$, $Y_i = \frac{Y_i}{\|Y_i\|_2}$. To extract the nonlinear features from multi-modal data, we use radial basis function (RBF) kernel mapping to generate kernel features. To be specific, we randomly choose $d$ samples from each modality as anchors $a_i^v$ and use the Gaussian kernel function to obtain the nonlinear features. Therefore, the kernel features of the $v$-th modality can be represented as $X_i^v = [exp(\frac{\|o_i^v - a_1^v\|_2^2}{-2\sigma^2}), exp(\frac{\|o_i^v - a_2^v\|_2^2}{-2\sigma^2}), \cdots, exp(\frac{\|o_i^v - a_d^v\|_2^2}{-2\sigma^2})]^T$, where $\sigma$ is the kernel width.

## 3.2 Formulation

CMH aims to learn compact hash codes while maintaining the inherent similarities of the original features in Hamming space. As is known to us all, learning consensus discriminative hash codes from multi-modal data has a significant challenge. Most CMH methods mainly emphasize leveraging the consistency of multi-modal data to learn consensus hash codes, which largely ignore modality-specific properties. Consequently, these methods cannot comprehensively explore the underlying data distribution among different modalities. To this end, we first employ modality-specific representations $B^v$ to capture the private property of each modality by the following formula.

$$\min_{B^v, W^v} \sum_{v=1}^{V} \|B^v - W^v X^v\|_F^2 \quad (1)$$
$$s.t. \ (W^v)^\top W^v = I, (B^v)^\top 1 = 0, (B^v)^\top B^v = nI,$$

where $W^v$ is the independent orthogonal projection matrix. In Eq.1, we adopt the bit decorrelation and balance constraints ($i.e.$, $(B^v)^\top 1 = 0$, $(B^v)^\top B^v = nI$) to encourage the compactness of modality-specific representations. Considering that multi-modal data encompasses information from the same instance, there should exist some shared information between different modalities. Further, we propose consensus hashing learning to excavate the common semantic information from modality-specific representations. Specifically, we preserve the similarities between the modality-specific properties and consensus hash codes $H$ from different modalities. Mathematically, the above problem can be represented as

$$\min_{B^v, W^v, H} \sum_{v=1}^{V} \|B^v - W^v X^v\|_F^2 + \beta \|H^\top B^v - rY^\top Y\|_F^2 \quad (2)$$
$$s.t. \ (W^v)^\top W^v = I, H \in \{-1, 1\}^{r \times n},$$
$$(B^v)^\top 1 = 0, (B^v)^\top B^v = nI,$$

where $\beta$ is the trade-off parameter.

In Eq.2, we can extract the common semantic information from modality-specific representations, thereby obtaining the consensus hash codes. As shown in Fig.1, multi-modal data from different modalities have different latent distributions. However, due to the lack of any prior information, orthogonal projection matrices could not be able to extract accurate modality-specific properties from multi-modal data. In other words, the learned specific and consensus representations lack the guidance of consistent information, thereby leading to deviation from the real data distribution. Naturally, we consider that each modality should adhere to a unified class center consistency criterion in order to mitigate disparities within latent distributions. Afterward, we propose distribution consistency

learning to ensure that specific and consensus representations adhere to the same class centers as the original data. Leveraging this concealed prior information, consensus hash codes can tap into a richer reservoir of underlying consensus data. Mathematically, we denote the problem as

$$\min_{U, F^v} \sum_{v=1}^{V} \|B^v - UF^v\|_F^2 + \|H - UE\|_F^2 \quad (3)$$
$$s.t. \ U^\top U = I, F^v(F^v)^\top = I, EE^\top = I.$$

where $F^v \in \mathbb{R}^{c \times n}$ and $E \in \mathbb{R}^{c \times n}$ are the encoding matrices, and $U \in \mathbb{R}^{l \times c}$ is the expected class center. We can observe that these matrices are orthogonal, which means each column of their matrices is independent. In essence, it can identify the most representative points to serve as class centers, thereby guiding hashing learning to aggregate iteratively hash codes around each class center.

Theoretically, distribution consistency learning offers the following two advantages: (1) it ensures that the projection matrices effectively map the original data closer to their respective class centers, thus mitigating the differences within latent distributions; (2) specific and consensus representations share the same class centers, which can sufficiently excavate the intrinsic correlation among each modality to improve the quality of hash codes.

In general, we can obtain the following unified objective function, $i.e.$,

$$\min_{\substack{B^v, W^v, U, \\ F^v, H}} \sum_{v=1}^{V} \|B^v - W^v X^v\|_F^2 + \alpha(\|B^v - UF^v\|_F^2$$
$$+ \|H - UE\|_F^2) + \beta \|H^\top B^v - rY^\top Y\|_F^2 \quad (4)$$
$$s.t. \ U^\top U = I, F^v(F^v)^\top = I, EE^\top = I,$$
$$(B^v)^\top 1 = 0, (B^v)^\top B^v = nI,$$
$$(W^v)^\top W^v = I, H \in \{-1, 1\}^{r \times n},$$

where $\alpha$ and $\beta$ are the trade-off parameters.

## 3.3 Optimization

In this section, we adopt the alternating solution strategy to optimize our proposed objection function. Specifically, we optimize one variable in each iteration while fixing other variables, and repeat all steps iteratively.

▶ $B^v$-**Step:** Fixing the irrelevant variables except $B^v$, we can solve the optimization problem, $i.e.$,

$$\min_{B^v} \sum_{v=1}^{V} \|B^v - W^v X^v\|_F^2 + \alpha \|B - UF^v\|_F^2$$
$$+ \beta \|H^\top B^v - rY^\top Y\|_F^2 \quad (5)$$
$$s.t. \ (B^v)^\top 1 = 0, (B^v)^\top B^v = nI.$$

Then, we can transform Eq.5 as the trace form, $i.e.$,

$$\max_{B^v} tr(J^v(B^v)^\top) \ s.t. \ (B^v)^\top 1 = 0, (B^v)^\top B^v = nI, \quad (6)$$

where $J^v = W^v X^v + \alpha UF^v + \beta rHY^\top Y$. We first define $\Delta = I_n - \frac{1}{n}1_n1_n^\top$. Then, we compute the singular value decomposition (SVD) of $J^v \Delta (J^v)^\top$ as follows

$$J^v \Delta (J^v)^\top = \left[ S^v | \overline{S}^v \right] \left[ \begin{array}{c|c} \Lambda^v & 0 \\ \hline 0 & 0 \end{array} \right] \left[ S^v | \overline{S}^v \right]^\top \quad (7)$$

where $\Lambda^v \in \mathbb{R}^{r^* \times r^*}$ and $S^v \in \mathbb{R}^{n \times r^*}$. Note here that $r^*$ is the rank of $J^v \Delta (J^v)^\top$. $\overline{S}^v$ is a matrix of the eigenvectors corresponding to zero eigenvalues from the remaining $r-r^*$ number of eigenvectors. Then, we adopt Gram-Schmidt process on $\overline{S}^v$ to obtain $\hat{S}^v \in \mathbb{R}^{n \times (r-r^*)}$. Afterwards, we define $Q^v = \Delta(Q^v)^\top S^v (\Lambda^v)^{-1/2} \in \mathbb{R}^{r \times r^*}$ and the random matrix $\hat{Q}^v \in \mathbb{R}^{r \times (r-r^*)}$. Finally, according to [35], we can obtain the optimal solution of $B^v$ as follows

$$B^v = \sqrt{n} \left[ S^v | \hat{S}^v \right] \left[ Q^v | \hat{Q}^v \right]^\top \qquad (8)$$

When $r = r^*$, $\hat{S}^v$, $\overline{S}^v$, and $\hat{Q}^v$ are empty.

▶ **H-Step:** Dropping the irrelevant variables, the $H$-subproblem can be expressed as

$$\min_H \sum_{v=1}^{V} \alpha \|H - UE\|_F^2 + \beta \|H^\top B^v - rY^\top Y\|_F^2 \qquad (9)$$
$$s.t. \ H \in \{-1, 1\}^{r \times n}.$$

Equivalently, we can transform Eq.9 into the following problem

$$\max_H tr(( \sum_{v=1}^{V} (\alpha UE + \beta r B^v Y^\top Y)) H^\top). \qquad (10)$$

Therefore, the solution can be obtained as follows

$$H = sgn(\sum_{v=1}^{V} (\alpha UE + \beta r B^v Y^\top Y)), \qquad (11)$$

where $sgn(\cdot)$ represents the element-wise indicator operator.

▶ **$W^v$-Step:** To update $W^v$, we fix the other variables to solve the subproblem as follows

$$\min_{W^v} \sum_{v=1}^{V} \|B^v - W^v X^v\|_F^2 \ s.t. \ (W^v)^\top W^v = I. \qquad (12)$$

Then, we adopt the trace operator to replace Eq.12. Thus, we can obtain

$$\max_{(W^v)^\top W^v = I} tr((W^v X^v)^\top B^v)$$
$$= \max_{(W^v)^\top W^v = I} tr((W^v)^\top B^v (X^v)^\top) \qquad (13)$$

We compute the singular value decomposition (SVD) on $B^v(X^v)^\top$. Therefore, $W^v$ can be solved by $M_w^v (N_w^v)^\top$, where $M_w^v$ and $N_w^v$ are the left and right singular values, respectively.

▶ **U-Step:** To obtain $U$, the subproblem can be rewritten as

$$\min_U \sum_{v=1}^{V} \|B^v - UF^v\|_F^2 + \|H - UE\|_F^2 \qquad (14)$$
$$s.t. \ U^\top U = I.$$

Similar to problem 1, we have

$$\max_{(U)^\top U = I} tr(U^\top (\sum_{v=1}^{V} B^v (F^v)^\top + HE^\top)) \qquad (15)$$

Therefore, the optimal solution can be computed by SVD, i.e., $M_u^v (N_u^v)^\top$.

▶ **$F^v$-Step:** To update $F^v$, the Eq.5 can be formed as

$$\min_{F^v} \sum_{v=1}^{V} \|B^v - UF^v\|_F^2 \ s.t. \ F^v (F^v)^\top = I. \qquad (16)$$

Similar to problem 1, we have

$$\max_{F^v (F^v)^\top = I} tr\left(F^v)^\top (U^\top (B^v)^\top)\right) \qquad (17)$$

Then, we employ SVD to compute the subproblem, i.e., $M_f^v (N_f^v)^\top$.

▶ **E-Step:** We drop irrelevant terms and obtain the following simplified problem

$$\min_E \sum_{v=1}^{V} \|H - UE\|_F^2 \ s.t. \ EE^\top = I. \qquad (18)$$

Hence, similar to problem 1, we can obtain the solution by SVD, i.e., $M_e (N_e)^\top$.

## 3.4 Out-of-Sample Extension

Once we generate the consensus hash codes, we can further learn hash function to achieve CMR for new queries. Specifically, we can obtain modality-specific hash function by linear regression, i.e.,

$$\min_{P^v} \sum_{v=1}^{V} \|H - P^v X^v\|_F^2 + \lambda \|P^v\|_F^2, \qquad (19)$$

where $\lambda$ is the trade-off parameter that avoids the trivial solution. Afterwards, we can directly optimize hash function $P^v$ as

$$P^v = H(X^v)^\top (X^v (X^v)^\top + \lambda I)^{-1}. \qquad (20)$$

Then, to achieve out-of-sample extension, we map all query samples into binary codes as follows

$$H_{query}^v = sgn(P^v K^v), \qquad (21)$$

where $K^v$ is the kernelized features of query samples.

## 3.5 Complexity Analysis

Our proposed DCGH mainly contains the computational complexity of each subproblem. To be specific, the computational complexity includes $O(Vldn+Vldn+cln+c^2ln)$ for solving $B^v$, $O(Vlcn+Vcln+Vc^2ln+ln)$ for solving $H$, $O(ldn+l^2d)$ for solving $W^v$, $O(2Vlcn+l^2n)$ for solving $U$, $O(Vldn+l^2n)$ for solving $F^v$, $O(Vldn+l^2n)$ for solving $E$, respectively. For computing hash function $P^v$, the complexity is about $O(d^2 + d^2l + d^2n + dln)$. Due to $V, d, c, l \ll n$, the total complexity of our method is approximately $O(n)$.

## 4 EXPERIMENTS

### 4.1 Datasets

Four widely used benchmark datasets are meticulously selected and preprocessed to fulfill the requirements of our experiments, including WIKI [15], MIRFlickr-25K [8], IAPR-TC12 [4], and NUS-WIDE [3]. We randomly choose 693 data pairs from WIKI to serve as the query set, while 1867 pairs from NUS-WIDE. For the other two datasets, we randomly choose 2,000 pairs from each as the query set. The remaining portions of the four datasets constitute the training set. The detailed dataset statistics could be found in the supplementary materials.

**Table 1: The mAP results (%) with different lengths of hash codes on four fully paired data.**

| Task | Method | WIKI | | | | MIRFlickr-25K | | | | IAPR-TC12 | | | | NUS-WIDE | | | |
|------|--------|------|------|------|------|------|------|------|------|------|------|------|------|------|------|------|------|
| | | 8 | 16 | 32 | 64 | 8 | 16 | 32 | 64 | 8 | 16 | 32 | 64 | 8 | 16 | 32 | 64 |
| Image ↓ Text | RFDH [23] | 21.96 | 22.89 | 22.69 | 22.98 | 58.25 | 58.46 | 58.28 | 58.22 | 35.32 | 44.85 | 45.53 | 45.83 | 34.54 | 47.33 | 57.76 | 58.32 |
| | LCMFH [21] | 29.88 | 32.35 | 32.10 | 33.83 | 67.44 | 69.86 | 70.88 | 69.49 | 32.69 | 42.73 | 44.70 | 45.69 | 55.52 | 63.18 | 64.21 | 64.87 |
| | MTFH [12] | 25.68 | 32.67 | 33.20 | 33.25 | 65.82 | 72.45 | 73.12 | 73.64 | 47.14 | 48.32 | 50.45 | 51.98 | / | / | / | / |
| | FCMH [28] | 26.47 | 31.00 | 31.42 | 31.81 | 72.59 | 73.69 | 75.20 | 75.15 | 46.25 | 49.48 | 51.70 | 53.20 | 64.66 | 65.82 | 66.40 | 67.20 |
| | FDDH [13] | 28.95 | 33.18 | 34.00 | 35.16 | 70.38 | 72.97 | 73.33 | 75.81 | 44.37 | 48.04 | 52.29 | 53.89 | 59.75 | 62.07 | 65.79 | 68.60 |
| | BATCH [29] | 31.73 | 34.37 | 36.93 | 38.16 | 71.63 | 73.31 | 73.69 | 74.10 | 44.91 | 48.05 | 50.40 | 52.62 | 63.17 | 65.72 | 66.49 | 67.47 |
| | EDMH [1] | 30.65 | 32.20 | 34.23 | 36.71 | 71.18 | 73.22 | 73.87 | 74.00 | 46.39 | 49.86 | 50.85 | 52.43 | 64.56 | 65.83 | 67.16 | 67.44 |
| | DAH [37] | 32.47 | 35.21 | 37.09 | 38.66 | 70.32 | 72.33 | 72.46 | 72.63 | 43.40 | 44.72 | 48.15 | 52.01 | 62.63 | 63.58 | 66.29 | 66.31 |
| | ALECH [10] | 30.28 | 33.46 | 34.95 | 36.98 | 71.95 | 73.54 | 74.00 | 74.28 | 45.68 | 48.30 | 50.35 | 52.07 | 65.02 | 66.08 | 67.85 | 68.22 |
| | WASH [35] | 32.76 | 33.43 | 36.97 | 37.45 | 71.18 | 72.53 | 72.72 | 73.03 | 46.75 | 48.25 | 51.00 | 53.45 | 62.45 | 64.04 | 64.18 | 63.34 |
| | AMSH [14] | 32.53 | 33.71 | 35.95 | 37.34 | 72.56 | 73.78 | 74.29 | 74.89 | 46.81 | 49.05 | 51.62 | 53.66 | 64.63 | 65.37 | 67.60 | 67.34 |
| | Our DCGH | **33.84** | **37.35** | **38.84** | **39.87** | **74.30** | **75.73** | **75.84** | **76.75** | **49.55** | **52.30** | **54.68** | **56.23** | **66.72** | **68.23** | **69.00** | **69.26** |
| Text ↓ Image | RFDH [23] | 46.24 | 51.23 | 54.10 | 54.53 | 58.64 | 57.66 | 57.97 | 57.85 | 34.83 | 45.52 | 46.40 | 57.54 | 35.48 | 53.66 | 58.22 | 62.73 |
| | LCMFH [21] | 66.08 | 69.46 | 72.71 | 71.38 | 70.93 | 74.48 | 74.65 | 74.15 | 34.69 | 49.86 | 53.68 | 56.42 | 58.43 | 67.08 | 72.23 | 73.64 |
| | MTFH [12] | 56.27 | 70.44 | 72.37 | 73.83 | 69.42 | 79.44 | 81.73 | 80.24 | 52.27 | 57.36 | 60.92 | 62.33 | / | / | / | / |
| | FCMH [28] | 65.02 | 65.98 | 67.73 | 67.82 | 79.76 | 81.75 | **83.57** | 83.69 | 53.47 | 58.50 | 61.92 | 65.13 | 75.57 | 77.64 | 78.84 | 80.76 |
| | FDDH [13] | 71.43 | 73.01 | 73.28 | 74.61 | 74.53 | 78.09 | 79.45 | 82.54 | 49.33 | 55.16 | 61.14 | 65.00 | 70.20 | 74.79 | 77.98 | 81.58 |
| | BATCH [29] | 71.67 | 72.75 | 74.90 | 76.65 | 79.01 | 80.65 | 81.35 | 82.05 | 52.75 | 57.77 | 61.85 | 64.88 | 76.57 | 77.58 | 79.41 | 80.20 |
| | EDMH [1] | 65.90 | 67.84 | 67.30 | 70.82 | 79.59 | 81.53 | 82.61 | 83.20 | 53.61 | 58.70 | 60.53 | 63.53 | 73.12 | 78.50 | 79.61 | 79.64 |
| | DAH [37] | 70.34 | 73.42 | 75.30 | 76.03 | 77.47 | 79.20 | 81.03 | 81.55 | 49.80 | 54.75 | 58.17 | 61.17 | 73.82 | 77.45 | 78.05 | 79.09 |
| | ALECH [10] | 70.52 | 74.43 | 74.53 | 74.81 | 78.06 | 80.75 | 81.73 | 82.15 | 52.55 | 57.74 | 61.44 | 64.61 | 76.26 | 77.64 | 78.89 | 79.77 |
| | WASH [35] | 70.65 | 74.86 | 75.40 | 75.59 | 76.69 | 78.53 | 79.59 | 79.77 | 50.89 | 54.25 | 61.50 | 65.02 | 73.31 | 77.70 | 80.39 | 81.09 |
| | AMSH [14] | 71.87 | 72.03 | 71.47 | 72.89 | 80.12 | 81.69 | 82.90 | 82.86 | 53.89 | 58.87 | 62.98 | 66.32 | 77.05 | 78.46 | 80.12 | 80.83 |
| | Our DCGH | **75.10** | **75.78** | **76.89** | **77.63** | **80.55** | **83.01** | 83.52 | **84.19** | **55.94** | **60.95** | **64.75** | **67.29** | **77.77** | **80.32** | **80.99** | **81.74** |

## 4.2 Baselines

To evaluate the performance of our DCGH, we conduct some experiments for 11 state-of-the-art CMH methods, including RFDH [23], LCMFH [21], MTFH [12], FCMH [28], FDDH [13], BATCH [29], EDMH [1], DAH [37], ALECH [10], WASH [35] and AMSH [14]. For the fairness, the parameters of baseline methods are consistent with those in the original literature. For the hyper-parameters (*i.e.*, $\alpha$, $\beta$, $\lambda$), we set them as $\{10^1, 10^1, 10^{-2}\}$, $\{10^4, 10^{-1}, 10^{-2}\}$, $\{10^3, 10^{-2}, 10^{-3}\}$, and $\{10^4, 10^{-1}, 10^{-4}\}$ on the previous mentioned four datasets, respectively. Moreover, the number of anchors in the RBF operation is configured as 1500, while the maximum iteration number is set to 10.

## 4.3 Experimental Setup and Evaluation Metrics

For ease of expression, we use the image and text modalities as an illustrative example to perform CMR tasks. To evaluate the performance of our proposed DCGH, we choose two typical CMR tasks: using visual data to search relevant textual data (*i.e.*, Image → Text task) and using textual data to search relevant visual data (*i.e.*, Text → Image task). Besides, we employ mean Average Precision (mAP) and Precision-Recall (PR) curves as evaluation metrics for the two retrieval tasks. In practical retrieval scenarios, cross-modal data is often collected from image-text pairs on the Internet, which inevitably leads to some mismatched pairs, *i.e.*, unpaired data. In our experiments, we randomly shuffle the first 25% of image data and the last 25% of text data to simulate the situation with 50%

unpaired data. Therefore, considering the prevalence of unpaired data in practical retrieval scenarios, we evaluate the performance of DCGH with both fully paired and partially paired settings on four datasets.

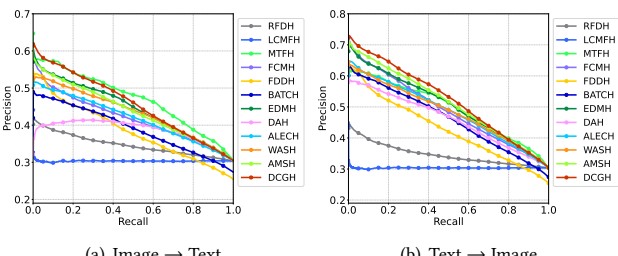

(a) Image → Text    (b) Text → Image

**Figure 2: PR curves with 8 bits on IAPR-TC12.**

## 4.4 Comparison on Fully Paired Datasets

In the scenario where data is fully paired, we adjust the hash lengths from 8 to 64 (*i.e.*, 8, 16, 32, 64 bits) to assess the performance of our DCGH. Table 1 showcases the mAP results of DCGH and other baselines with different hash lengths on four datasets. Additionally, Fig.2 and Fig.3 illustrate the PR and Top-k precision curves for various methods with 8-bit hash codes on four datasets. It is imperative to highlight that the MTFH method cannot execute on the NUS-WIDE dataset due to memory overflow. According to these results, we can obtain the following observations: (1) The

Table 2: The mAP results (%) with different lengths of hash codes on three partially paired datasets.

| Task | Method | MIRFlickr-25K | | | | IAPR-TC12 | | | | NUS-WIDE | | | |
|------|--------|----|----|----|----|----|----|----|----|----|----|----|----|
| | | 8 | 16 | 32 | 64 | 8 | 16 | 32 | 64 | 8 | 16 | 32 | 64 |
| Image ↓ Text | RFDH [23] | 57.08 | 56.91 | 57.20 | 57.45 | 33.33 | 32.53 | 33.00 | 33.04 | 35.90 | 36.58 | 36.89 | 37.08 |
| | LCMFH [21] | 67.24 | 67.03 | 68.16 | 69.08 | 35.42 | 36.02 | 36.79 | 38.54 | 60.40 | 63.27 | 65.06 | 65.03 |
| | MTFH [12] | 65.52 | 71.82 | 72.14 | 73.15 | 30.16 | 28.48 | 28.65 | 27.71 | / | / | / | / |
| | FCMH [28] | 70.32 | 73.26 | 73.54 | 74.26 | 47.28 | 48.78 | 50.95 | 52.95 | 62.98 | 63.51 | 64.57 | 65.89 |
| | FDDH [13] | 69.79 | 71.68 | 73.14 | 75.72 | 41.98 | 47.25 | 50.39 | 53.07 | 61.79 | 63.67 | 66.82 | 67.08 |
| | BATCH [29] | 62.85 | 63.83 | 64.29 | 64.40 | 45.30 | 47.61 | 49.81 | 51.74 | 63.63 | 65.23 | 66.23 | 67.28 |
| | EDMH [1] | 70.09 | 72.67 | 73.33 | 73.66 | 47.80 | 47.60 | 50.20 | 50.91 | 64.23 | 64.80 | 66.65 | 67.36 |
| | DAH [37] | 69.25 | 71.45 | 72.12 | 71.78 | 40.02 | 43.63 | 46.83 | 49.94 | 59.51 | 62.90 | 66.24 | 66.38 |
| | ALECH [10] | 70.77 | 72.27 | 72.56 | 73.17 | 44.46 | 46.78 | 49.17 | 50.84 | 64.57 | 66.53 | 67.04 | 67.43 |
| | WASH [35] | 71.36 | 72.13 | 72.61 | 72.94 | 45.79 | 47.83 | 49.17 | 50.84 | 63.99 | 65.03 | 64.75 | 64.53 |
| | AMSH [14] | 71.82 | 73.17 | 73.89 | 74.12 | 46.38 | 48.25 | 50.50 | 52.41 | 64.04 | 65.12 | 66.42 | 67.75 |
| | Our DCGH | **74.33** | **75.63** | **74.93** | **76.16** | **48.72** | **51.59** | **53.77** | **55.27** | **66.44** | **66.71** | **67.30** | **68.17** |
| Text ↓ Image | RFDH [23] | 57.23 | 57.37 | 57.44 | 57.61 | 33.95 | 33.89 | 34.78 | 35.41 | 35.53 | 36.57 | 36.66 | 36.83 |
| | LCMFH [21] | 71.15 | 71.35 | 73.40 | 74.85 | 36.03 | 37.54 | 39.78 | 40.23 | 71.28 | 74.24 | 76.46 | 76.39 |
| | MTFH [12] | 75.60 | 79.67 | 80.33 | 81.35 | 52.36 | 56.46 | 60.91 | 63.27 | / | / | / | / |
| | FCMH [28] | 78.06 | 80.48 | 81.47 | 82.22 | 52.79 | 56.86 | 59.90 | 63.04 | 74.63 | 74.95 | 75.43 | 76.17 |
| | FDDH [13] | 73.29 | 76.12 | 78.90 | 81.85 | 46.01 | 53.92 | 59.14 | 63.41 | 73.48 | 76.61 | 78.89 | 79.80 |
| | BATCH [29] | 78.65 | 78.99 | 80.70 | 81.44 | 51.51 | 56.56 | 60.71 | 63.59 | 72.52 | 75.66 | 77.58 | 78.89 |
| | EDMH [1] | 78.04 | 80.12 | 81.04 | 81.81 | 52.31 | 55.20 | 58.74 | 60.55 | 76.29 | 78.09 | 79.93 | 79.86 |
| | DAH [37] | 76.64 | 77.00 | 78.96 | 80.16 | 48.88 | 53.43 | 55.39 | 58.67 | 74.35 | 77.37 | 77.31 | 79.07 |
| | ALECH [10] | 77.91 | 79.54 | 80.71 | 81.12 | 52.08 | 56.43 | 60.22 | 63.42 | 75.37 | 77.76 | 78.60 | 79.04 |
| | WASH [35] | 76.88 | 78.46 | 78.94 | 79.49 | 51.11 | 55.25 | 58.06 | 60.52 | 72.81 | 74.81 | 75.99 | 76.79 |
| | AMSH [14] | 78.68 | 80.50 | 81.34 | 81.68 | 53.22 | 57.09 | 61.26 | 64.46 | 76.35 | 78.04 | 79.57 | 80.57 |
| | Our DCGH | **79.47** | **81.93** | **81.89** | **83.21** | **55.00** | **59.38** | **63.56** | **66.03** | **77.03** | **78.39** | **80.50** | **80.71** |

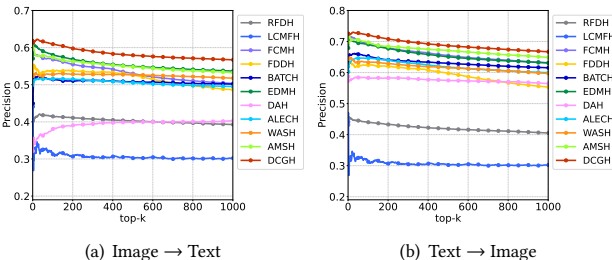

Figure 3: Top-k precision curves with 8 bits on IAPR-TC12.

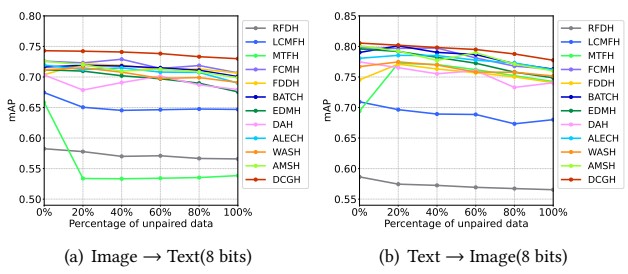

Figure 4: The mAP results with varying proportions of unpaired data with 8 bits on MIRFlickr-25K.

proposed DCGH outperforms almost all baselines on both Image → Text and Text → Image tasks on four datasets with different hash lengths. The results demonstrate the proposed modality-specific

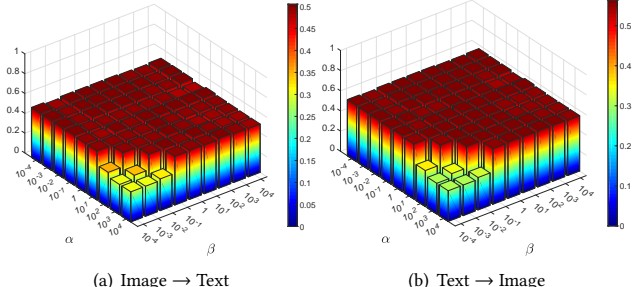

Figure 5: The mAP scores with 8 bits in terms of parameters $\alpha$ and $\beta$ on IAPR-TC12.

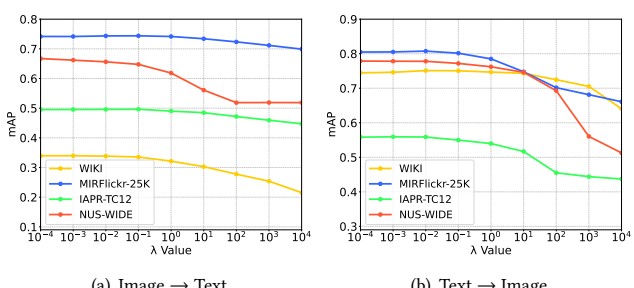

Figure 6: The mAP scores with 8 bits for $\lambda$ variations.

representation and distribution consistency learning can enhance

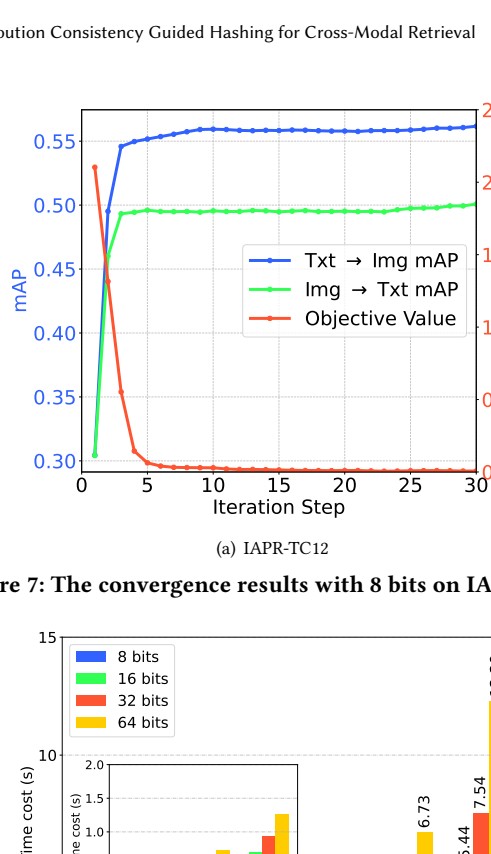

(a) IAPR-TC12

**Figure 7: The convergence results with 8 bits on IAPR-TC12.**

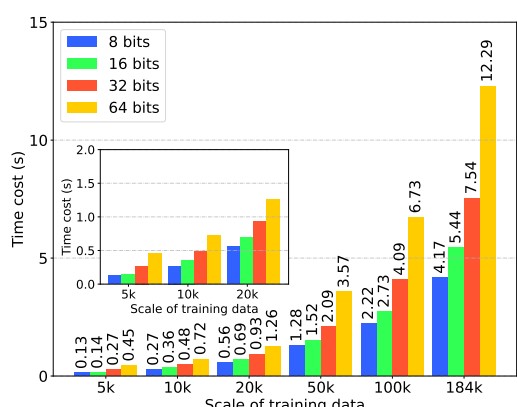

**Figure 8: Training time (second) of DCGH with different data scales on NUS-WIDE dataset.**

the quality of hash codes, thereby obtaining higher performance. (2) Generally, the performance of Text → Image tasks is better than that of Image → Text tasks. This could be attributed to the fact that features extracted from textual information contain more semantic information. (3) As the hash length increases, the mAP scores of all comparison methods also improve. This proves that longer hash codes are capable of encoding more semantic information. Furthermore, we find that DCGH achieves more significant performance improvements in cases with shorter hash code lengths. This indicates that the consensus hashing learning strategy used in our DCGH can learn more compact and discriminative hash codes. (4) By comparing the PR curves of various methods, we find that the coverage area of DCGH exceeds almost all baselines. Furthermore, the top-k precision curves illustrate that DCGH achieves nearly the highest precision across different k values. These observations underscore the superiority of DCGH in terms of query coverage and accuracy.

## 4.5 Comparison on Partially Paired Datasets

On three partially paired datasets, we adjust the code lengths from 8 to 64 bits to evaluate the performance of our DCGH. The mAP

scores of all methods are illustrated in Table 2. In addition, to show the performance under different proportions of unpaired data, we draw the corresponding mAP results on MIRFlickr-25K with 8 bits in Fig.4. Based on these results, we can obtain the following observations: (1) On three partially paired datasets, our DCGH still exhibits the highest performance, which demonstrates its high robustness against unpaired data. With the hash length increases, the mAP scores of all comparison methods also improve. (2) The performance of all methods decreases as the proportion of unpaired data increases. This could be attributed to the significant impact of the semantic information provided by labels on supervised learning. (3) Under all proportions of unpaired data and different code lengths on MIRFlickr-25K, our DCGH achieves minimal performance degradation. This could be attributed to our proposed distribution consistency learning strategy, which utilizes the latent class centers to alleviate the errors introduced by mismatched pairs. (4) Similar to the fully paired situation, the performance of all methods increases with the growth of code length, as longer code lengths can carry more semantic information.

## 4.6 Parameter Sensitivity Analysis

We investigate the sensitivity of our DCGH on four datasets, which involve three hyper-parameters, namely, $\alpha$, $\beta$, and $\lambda$.

**Parameter $\alpha$ and $\beta$** determine the contribution of distribution consistency learning and consensus hash learning, respectively. To analyze the parameters sensitivity, we adopt grid search and set the range of $\alpha$ and $\beta$ as $[10^{-4}, 10^4]$. The corresponding mAP scores are depicted in Fig.5. It is evident that DCGH consistently achieves superior results across most ranges, demonstrating the stability of our approach.

**Parameter $\lambda$** serves as the regularization coefficient in the hash function learning phase, and Fig.6 shows the mAP results of DCGH with different $\lambda$ values on four datasets. Clearly, within the range of $[10^{-4}, 10^{-1}]$, the mAP curves remain almost horizontal, indicating that the values of $\lambda$ are feasible within this range. In addition, we further discuss the impact of eliminating the $\lambda$ parameter on the performance of DCGH in the ablation analysis section.

## 4.7 Convergence Analysis

As previously mentioned, we can iteratively solve each sub-problem to obtain the optimal solution. In order to illustrate the convergence characteristics of DCGH more intuitively, we draw the convergence curves and mAP curves across four datasets in Fig.7. From the figure, we can make the following observations: (1) The objective values decline rapidly within the first five steps and manage to converge before 10-th step. (2) Similarly, the mAP values consistently increase with the progression of the iterative steps, attaining stable points around the 10-th step. These results demonstrate that our proposed DCGH boasts rapid and stable convergence advantages.

## 4.8 Comparison on Time Cost

To show the computational complexity of DCGH and its scalability on large datasets, we conduct experiments on NUS-WIDE with varying hash code lengths and data scales. The bit length extends from 8 to 64, and the data scale varies from 10K to 184K. All the results are depicted in Fig.8. The observation reveals that the time

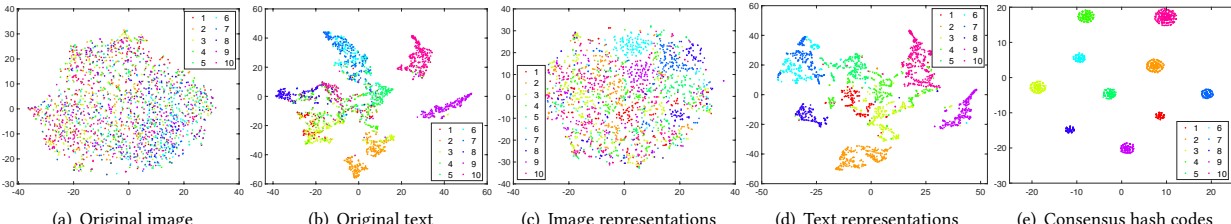

(a) Original image     (b) Original text     (c) Image representations     (d) Text representations     (e) Consensus hash codes

**Figure 9: t-SNE visualizations with 64 bits on the WIKI dataset.**

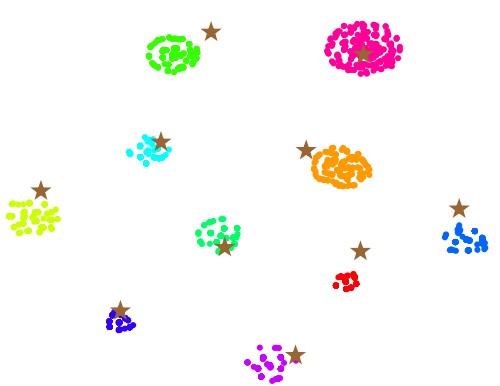

**Figure 10: Visualization of consensus hash codes and the corresponding class centers on the WIKI dataset. (Pentagrams represent class centers.)**

consumption of DCGH exhibits a linear-like relationship with both data scale and hash code length. More importantly, we notice that the training time remains within an impressive range even when the data scale exceeds 100K. This indicates the potential applicability of DCGH to larger multimedia datasets and practical application scenarios. Moreover, we compare the training time of different methods across four datasets in supplementary materials.

**Table 3: The ablation results (%) with 64 bits on four datasets.**

| Task | Method | WIKI | MIRFlickr-25K | IAPR-TC12 | NUS-WIDE |
|------|--------|------|---------------|-----------|----------|
| Image ↓ Text | DCGH-$d$ | 37.84 | 75.39 | 54.45 | 67.40 |
| | DCGH-$r$ | **40.04** | 54.06 | 34.48 | 52.12 |
| | DCGH-$c$ | 36.71 | 74.00 | 52.43 | 67.44 |
| | DCGH | 39.87 | **76.75** | **56.23** | **69.26** |
| Text ↓ Image | DCGH-$d$ | 76.37 | 83.19 | 65.88 | 79.74 |
| | DCGH-$r$ | 76.38 | 76.60 | 56.25 | 69.83 |
| | DCGH-$c$ | 70.82 | 83.20 | 63.53 | 79.64 |
| | DCGH | **77.63** | **84.19** | **67.29** | **81.74** |

### 4.9 Ablation Analysis

To further validate the effectiveness and necessity of the individual components in our proposed DCGH, we perform an ablation study on three variants of DCGH across four datasets. Specifically, DCGH-d removes the distribution consistency learning component represented in Eq.3, DCGH-r eliminates the regularization term, *i.e.*, the second term in Eq.19, and DCGH-c discards the consensus hashing learning strategy. It should be noted that discarding

the consensus hashing learning strategy implies abandoning the overarching DCGH framework. Hence, we utilize FCMH [28] as the substitute for DCGH-c in our study. Table 3 shows the mAP scores of DCGH and its variants across four datasets. Overall, the original DCGH consistently exhibits superior performance compared to its variant models in the majority of scenarios. However, the performance of DCGH-r drops most significantly because, without the constraint of the regularization term, the method can be susceptible to converge into a trivial solution. In addition, DCGH-d and DCGH-c also experience noticeable performance degradation, which proves the importance of distribution consistency learning and consensus hashing learning to DCGH, respectively.

### 4.10 Visualization Analysis

To provide an intuitive understanding for our proposed DCGH, we utilize t-SNE to visualize the original data, the learned modality-specific representations, and the final consensus hash codes on the WIKI dataset. Fig.9 (a) and (b) show the visualization results of the original image and text data. We can observe that the original data is chaotic and indistinguishable. Fig.9 (c) and (d) depict the spatial distribution of the two modality-specific representations. It can be observed that the specific representations from different categories start to exhibit a preliminary class structure. Fig.9 (e) represents the final consensus hash codes, which indicates our DCGH enjoys more discrimination and a more compact class structure. Moreover, we also adopt t-SNE to visualize the learned consensus hash codes and class centers as shown in Fig.10. We can observe that instances from the same category cluster around their respective class centers under the guidance of the class centers. This demonstrates the effectiveness of the proposed distribution consistency learning strategy.

## 5 CONCLUSION

In this paper, we propose a novel distribution consistency guided hashing (DCGH) for cross-modal retrieval, which can handle the cases of paired and unpaired. To overcome the heterogeneity of multi-modal data, we learn consistent hash codes from the extracted modality-specific representation, thereby endowing the learned hash codes more semantics with specific and consistent information. Moreover, we propose distribution consistency learning to project the original data closer to their respective class centers, thus mitigating the differences within latent distributions. Experimental results on four benchmarks demonstrate that our DCGH outperforms 11 state-of-the-art CMH methods.

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
