# OpenReview forum: "Distribution Consistency Guided Hashing for Cross-Modal Retrieval"
_acmmm.org/ACMMM/2024/Conference — MM2024 Poster_

### Official Review · Reviewer_id12 · 2024-04-28

**Rating:** 6
**Confidence:** 4

**Summary:**

The paper propose a Distribution Consistency Guided Hashing (DCGH) framework, which learns
the modality-specific representation to extract the private discriminative information and consensus hash codes from
the private representation by consensus hashing learning, thereby merging the specifics with consistency. Also, a distribution consistency learning  between multi-modal data is proposed. DCGH first extracts modality-specific representations to enhance the quality of private properties in each modality. It then employs consensus hashing learning to capture consistent structural information across all modalities. By ensuring a similar class distribution for both modality-specific representations and consensus hash codes, DCGH explores more consistent information for improved retrieval accuracy. The framework's contributions include simultaneously investigating modality-specific and distribution consistency learning, enabling a comprehensive exploration of correlations between modalities.Overall, DCGH offers a promising approach to CMR by balancing modality-specific details with consensus information, ultimately enhancing retrieval efficiency and performance for large-scale multi-modal datasets.

**Strengths:**

- English writing is easy to follow.
- Experiment is suffient.
- Theory proof is consummate.


In paper novelity, this paper has highlight such as:
- By extracting modality-specific representations, DCGH can capture the unique discriminative features present in each modality. This allows for a more detailed and nuanced representation of the data, improving the quality of the learned features.
- The inclusion of modality-specific representations helps in better capturing the inherent characteristics of each modality, leading to improved retrieval accuracy.

**Limitations:**

- The framework figure should be bigger.
- The citation is not suffient, more categories can be summaried in related work part.
- more categories can be summaried in related work part.
-  Some insignificant figure can place in supplementary material, such as Fig.7.

**Suitability:**

3

---

### Official Review · Reviewer_3Ezt · 2024-05-15

**Rating:** 6
**Confidence:** 3

**Summary:**

This paper presents a novel Distribution Consistency Guided Hashing (DCGH) for cross-modal retrieval. It adopts the modality-specific representation to extract the private discriminative information. Subsequently, DCGH learns consensus hash codes from the private representation by consensus hashing learning, thereby merging the specifics with consistency. Finally, the authors proposes distribution consistency learning to guide hash codes following a similar class distribution principle between multi-modal data, thereby exploring more consistent information.
Expensive experiments demonstrate the effectiveness of the proposed DCGH on both fully paired and partially paired cross-modal retrieval tasks.

**Strengths:**

(1) This work is interesting and novel, which proposes distribution consistency learning that makes each specific representation obey the same class centers as the original data.
(2) The paper is well written and the motivation is clear. The existing methods often rely on pairwise similarities to learn hash codes and ignore the spatial distribution correlations, which could not be able to fully mine consensus class information.
(3) Extensive experimental results show the DCGH outperforms state-of-the-art methods on the evaluation metrics. The analysis on distribution consistency learning is performed effectively.

**Limitations:**

(1) In Figure 2, authors should analyze the reason why DSFH seems to perform worse than AMSH on image retrieval text tasks.
(2) For unpaired data, the authors should analyze why the performance drop is not obvious under different ratios?
(3) The English presentation of the work needs to be improved. Such as '\lambda variations' in line 692.
(4) For Figure 6, what happens to the retrieval performance when the value of \lambda is small.

**Suitability:**

3

---

### Official Review · Reviewer_ohgK · 2024-05-20

**Rating:** 5
**Confidence:** 3

**Summary:**

This paper proposes a novel distribution consistency guided hashing (DCGH) for cross-modal retrieval, which can handle the cases of paired and unpaired. Specifically, DCGH first learns consistent hash codes from the extracted modality-specific representation, thereby endowing the learned hash codes more semantics with specific and consistent information. Then, distribution consistency learning is proposed that projects the original data closer to their respective class centers, thus mitigating the differences within latent distributions. Lots of experimental results show the effectiveness of DCGH.

**Strengths:**

1.The framework diagram and visualization results help the reader to effectively understand the algorithm motivation and flow.
2.The concepts of distribution consistency learning seem be interesting, and the paper is well-organized.
3. The work seems to be novel, and this paper has clear motivations and reasonable. It is the first to simultaneously investigate modality-specific and distribution consistency learning, which enables a comprehensive exploration of the inherent correlations between all modalities
4. The effectiveness and superiority are well demonstrated by experimental results with state-of-the-art CMH methods on four datasets with both fully paired and partially paired.

**Limitations:**

1.How the author judged the centers of distribution consistency.
2.The baselines exclude any deep learning hashing methods. The authors should compare some deep hashing methods.
3.There are some typos, such as the legend of Figure 7.
4.Some class center consistency methods have been adopted in some hashing methods. The authors should give some discussions about them. Such as Two-stream deep hashing with class-specific centers for supervised image search, Joint specifics and consistency hash learning for large-scale cross-modal retrieval.

**Suitability:**

3

---

### Official Review · Reviewer_2YwD · 2024-05-23

**Rating:** 5
**Confidence:** 4

**Summary:**

This paper proposes a novel distribution consistency guided hashing (DCGH) for cross-modal retrieval, which can handle the cases of paired and unpaired. The proposed distribution consistency learning aims to project the original data closer to their respective class centers, thus mitigating the differences within latent distributions. Experimental results on four benchmarks demonstrate that the proposed DCGH outperforms 11 state-of-the-art CMH methods. In general, this paper has clear motivation and is interesting.

**Strengths:**

- The structure of the paper is clear and well-written, which can make it easy for the readers to follow the proposed methodology.
- The idea is interesting and the proposed method is reasonable. The distribution consistency learning can make each specific representation to obey the same class centers as the original data, thereby exploring more consistent information.
- The experimental evaluation is impressive and convincing. A series of experimental results on four benchmarks demonstrate that the proposed DCGH outperforms 11 state-of-the-art CMH methods under the cases of paired and unpaired data.
- The proposed distribution consistency learning is novel, which can makes each specific representation obey the same class centers as the original data.

**Limitations:**

- For Figure 4, the authors should uniformly use the IAPR-TC12 dataset to present experimental results with different rates of unpaired data.
- The authors claim that the computational complexity of the proposed method is O(n), so the training time of all methods should be given for comparison.
- For t-SNE visualizations, the image representation has a more cluttered data structure than the text representation. The authors should further analyze it.
- For Figure 7, the legend seems to be wrong. What does Img mAP means?
- Authors should further analyze why different proportions of unpaired data has little impact on retrieval performance.

**Suitability:**

3

---

### Meta-Review · Area_Chair_x22E · 2024-07-01

**Recommendation:** Accept (Poster)
**Confidence:** 5

**Metareview:**

After rebuttal, the paper finally received four consistent positive recommendations. The reviewers agreed that the paper was innovative, well-experimented and convincing. I agree with the reviewers' suggestions and vote to accept the paper.